# IL-24 in COVID-19 Patients: Correlations with Disease Progression

**DOI:** 10.3390/ijms26178403

**Published:** 2025-08-29

**Authors:** Richard Vollenberg, Katharina Schütte-Nütgen, Markus Strauss, Jonel Trebicka, Julia Fischer, Phil-Robin Tepasse

**Affiliations:** 1Department of Medicine B for Gastroenterology, Hepatology, Endocrinology and Clinical Infectiology, University Hospital Muenster, 48149 Muenster, Germanyjulia.fischer2@ukmuenster.de (J.F.);; 2Department of Medicine D, Division of General Internal Medicine, Nephrology and Rheumatology, University Hospital of Münster, 48149 Muenster, Germany; 3Department of Cardiology, Faculty of Health, School of Medicine, University of Witten/Herdecke, 58455 Witten, Germany; 4Department of Cardiology I—Coronary and Peripheral Vascular Disease, Heart Failure Medicine, University Hospital Muenster, 48149 Muenster, Germany

**Keywords:** interleukin-24 (IL-24), COVID-19, disease severity, cytokines, apoptosis, immune response, acute respiratory distress syndrome (ARDS), biomarker, viral infections, inflammatory markers

## Abstract

Interleukin-24 (IL-24) is a cytokine known for its role in immune regulation and apoptosis, with potential implications in viral infections like COVID-19. This study aimed to investigate the association between IL-24 serum levels and the severity of COVID-19 disease. In this prospective bi-center cross-sectional study, we enrolled 41 COVID-19 patients from two hospitals in Germany. Serial blood samples were collected from a subset of patients, resulting in 88 total blood samples. Patients were categorized into critical, severe, moderate, and mild disease groups based on WHO criteria. IL-24 serum levels were measured during the acute or convalescent phase using an ELISA assay. Inflammatory markers, and kidney and liver function parameters were also evaluated. Statistical analysis included non-parametric tests and correlation analysis. Elevated IL-24 serum levels were observed in ambulant patients (mild disease), compared to hospitalized patients (critical, severe, moderate disease, *p* < 0.05). IL-24 levels were also significantly higher in patients without oxygenation disorder compared to those with oxygenation therapy (*p* < 0.05). A negative correlation was found between IL-24 levels and markers of inflammation and liver/kidney function. Elevated IL-24 serum levels were associated with milder COVID-19 courses, suggesting a protective role in modulating immune responses and promoting antiviral apoptosis. Conversely, reduced IL-24 in severe cases may reflect impaired immune regulation, highlighting its potential as a biomarker and therapeutic target.

## 1. Introduction

Severe acute respiratory syndrome coronavirus 2 (SARS-CoV-2) is responsible for the coronavirus disease 2019 (COVID-19, coronavirus disease 2019) pandemic. It can lead to severe pneumonia and respiratory failure [1]. While most individuals experience mild to moderate symptoms during viral infection, it can lead to a severe illness characterized by acute respiratory distress syndrome (ARDS) [2,3], resulting in substantial pulmonary damage and high mortality rates within a specific patient subgroup [4]. The severity of COVID-19 is associated with hyperinflammation resembling classic cytokine storm syndromes, causing progressive lung failure, and, in certain instances, leading to multiorgan failure and death [5]. Even mild cases can result in persistent symptoms post-infection [6]. The use of current SARS-CoV-2 vaccines can substantially reduce the risk of critical COVID-19 courses and death [7]. In contrast, protection against mild or asymptomatic infections is limited, partly due to viral evolution [8,9]. The combination of vaccination and prior infection (“hybrid immunity”) can induce a broader and more durable T-cell–mediated immune response. Unlike antibodies, T cells remain active across variants and play a central role in preventing severe disease courses—even when viral variants are able to partially evade antibody activity [10,11]. However, a substantial proportion of patients with immunodeficiency, certain underlying medical conditions (e.g., organ transplantation, hematologic malignancies, chronic lung diseases), or advanced age still remain insufficiently protected by COVID-19 vaccines against severe disease [12]. Therefore, additional therapeutic strategies such as host-based therapies are necessary for these patients [13].

Targeting apoptosis could be a promising strategy to limit viral infection, as it is part of the naturally acquired anti-viral mechanism. Interleukin-24 (IL-24)/MDA-7 is a protein of the interleukin-10 (IL-10) superfamily, which is expressed in hematopoietic and skin cells [14]. The cytokine was initially discovered in melanoma cells stimulated with IFN-β and mezerein [15]. Strikingly, it could be shown that IL-24 is the most upregulated differentially expressed genes in the lungs of H5 avian influenza virus-infected mice [16]. Furthermore, IL-24 exhibited antiviral effects in cells infected with influenza virus under in vivo conditions by inducing toll-like receptor 3 (TLR3)- mediated apoptotic pathways [17]. Taken together, IL-24 may selectively induce apoptosis in virus-infected cells [17]. Consistently, the administration of recombinant IL-24 led to an inhibited viral gene expression in vitro [18]. Moreover, upon SARS-CoV-2 infection TLRs such as TLR3, TLR7 and TLR 8 are also known to be activated [19]. However, the role of IL-24 in disease severity in patients with SARS-CoV-2 infection remains elusive.

Thus, the aim of this study is to investigate the association of IL-24 serum levels on the further course of SARS-CoV-2 infections in moderate, severe and critical infections.

## 2. Results

### 2.1. Patients Characteristics

In this prospective study, 41 COVID-19 patients (n = 88 blood samples) were enrolled. Among the hospitalized patients, 11 (37 serum samples) were critically ill, 9 (24 serum samples) were severely ill, and 8 (14 serum samples) had a moderate illness. Additionally, 13 outpatients were included after a mild course of the disease. There were no significant differences in terms of patient gender (*p* = 0.19), patient age (*p* = 0.314), and BMI (*p* = 0.057) among the patient groups. Blood samples (first sample) were collected from hospitalized patients at a median of 12 days (IQR 11–16) after symptom onset for critically ill patients, 11 days (IQR 3–17) for severely ill patients, and 8 days (IQR 2–8) for moderately ill patients.

Blood collection for outpatient patients after mild disease course (during the convalescent phase) occurred at a median of 38 days (36–46) after symptom onset (*p* < 0.001). Pre-existing conditions—including diabetes, chronic inflammatory bowel disease, respiratory diseases, neoplasm, and arterial hypertension—stratified by COVID-19 disease severity are shown in Table 1.

Hospitalized patients with acute disease showed increased inflammatory laboratory parameters (ferritin, interleukin-6, leukocytes, lymphocytes and C-reactive protein) compared to convalescent patients (Table 2). Critically ill COVID-19 patients exhibited elevated transaminases and gamma-GT values (Table 2).

### 2.2. IL-24 Serum Levels in COVID-19 Patients by Disease Severity and Sampling Time: Higher Proportion Above Lower Limit of Quantification in Mild and Moderate Cases

For all measurements below the lower limit of quantification (LLoQ; 62.5 pg/mL), values were imputed as LLoQ/2 (31.25 pg/mL). IL-24 serum concentrations were assessed in 41 COVID-19 patients (88 samples) across critical, severe, moderate, and mild disease courses (Table 3). At hospital admission or inpatient transfer (H1), median IL-24 levels were 31.3 pg/mL (IQR 31.3–31.3) in critical disease (n = 11, median time from symptom onset: 12 days (IQR 11–17)), 31.3 pg/mL (31.3–31.3) in severe disease (n = 9, 11 days (3–17)), and 31.3 pg/mL (31.3–661) in moderate disease (n = 8, 8 days (2–8)). The proportion of samples above the LLOQ was 9%, 0%, and 25%, respectively. On day 3 of hospitalization (H2), medians were 31.3 pg/mL (31.3–31.3) in critical disease (n = 9, 17 days (14–21)) and severe disease (n = 7, 14 days (4–22)); moderate disease (n = 2, 4 days) showed a median of 754.1 pg/mL (31.3). Above-LLOQ levels occurred in 11%, 0%, and 50% of cases, respectively. By day 8 (H3), IL-24 remained at 31.3 pg/mL (31.3–31.3) in both critical (n = 7, 22 days (17–26)) and severe (n = 5, 13 days (11–26)) disease, while the single moderate case (n = 1, 8 days) showed 1660 pg/mL. On day 14 (H4), critical disease (n = 4, 31 days (27–37)) had a median of 31.3 pg/mL (31.3–398.6), and the single moderate case (n = 1, 10 days) measured 1188 pg/mL. By day 21 (H5), only critical cases (n = 3, 46 days) were sampled, all showing 31.3 pg/mL (31.3–31.3) (Table 3, Figure 1).

A subset of these hospitalized patients was also included in the outpatient follow-up group (O2), representing the same individuals sampled after hospital discharge in our outpatient clinic 4–5 months after symptom onset. Medians were 31.3 pg/mL (31.3–31.3) in critical disease (n = 3, 140 days), 31.3 pg/mL (31.3–31.3) in severe disease (n = 3, 126 days), and 558 pg/mL (178) in moderate disease (n = 2, 137 days).

In the outpatient convalescent group with mild disease (O1) (n = 13, 38 days (36–46)), median IL-24 was 174 pg/mL (31.3–561), with 62% of samples above the LLOQ.

Across all time points, values above the LLOQ were numerical more frequently observed in moderate or mild disease, particularly among patients without oxygenation disorders, compared to those with severe or critical disease courses.

### 2.3. Comparison of IL-24 Serum Levels and Symptom-to-Sampling Times in Hospitalized vs. Convalescent Outpatient COVID-19 Patients

IL-24 serum concentrations and the interval from symptom onset to blood collection were compared between hospitalized COVID-19 patients (including critical, severe, and moderate disease courses) at the time of their first blood draw after hospital admission or inpatient transfer (H1), and outpatients after a mild disease course during convalescence (O1). For all IL-24 measurements below the lower limit of quantification (LLOQ; 62.5 pg/mL), values were set to half the LLOQ (31.25 pg/mL). Median IL-24 levels were significantly higher in outpatients compared to hospitalized patients (174 pg/mL (IQR 32.25–561) vs. 32.25 pg/mL (IQR 32.25–32.5), *p* = 0.007, Mann–Whitney U-test). The proportion of samples above the LLOQ was likewise greater in the outpatient group (62% vs. 11%). The median time from symptom onset to sampling differed markedly between the two groups, with outpatients sampled at a much later stage of the disease course (38 days (IQR 36–46)) compared to hospitalized patients at H1 (11 days (IQR 5–15); *p* < 0.001, Mann–Whitney U-test) (Figure 2).

### 2.4. Reduced IL-24 Serum Levels in COVID-19 Patients Are Associated with Indicators of Liver-/Kidney Damage and Lymphocytopenia

In the linear mixed-effects model, IL-24 levels exhibited a significant positive correlation with relative lymphocyte counts (β = 3.127, 95% CI [0.284, 5.971], *p* = 0.032) and absolute lymphocyte count (β = 33.661, 95% CI [4.177, 63.146], *p* = 0.026). Creatinine (β = 70.691, 95% CI [2.479, 138.9], *p* = 0.043) and gamma-GT (β = 0.396, 95% CI [0.154, 0.638], *p* = 0.002) demonstrated a negative significant correlation with IL-24 serum levels (Figure 3a–c).

## 3. Discussion

In our study, an association was observed between IL-24 serum levels and both the severity of COVID-19 disease and the time course of the illness.

IL-24, encoded on chromosome 1q32–33 and a member of the IL-10 gene family [20], is secreted by various immune cells including monocytes, T cells, dendritic cells [21], and skin cells [22]. IL-24 has known impacts on the immune system, chronic inflammatory diseases, skin homeostasis, and pregnancy [21,23], and it appears to play a unique role in promoting apoptosis specifically in infected or damaged cells [17,18,20].

The majority of COVID-19 patients in our study, with critical or severe disease (oxygenation impairment) showed no IL-24 concentrations above the LLOQ during their hospital stay, as determined from serial blood samples collected at time points H1–H5. In contrast, in patients with a moderate disease course (hospitalized without oxygenation impairment) and in outpatients in convalescence after a mild disease course, markedly elevated IL-24 levels were observed in some individuals at various observation time points (moderate disease: H1–H4; mild disease: O1). Compared to hospitalized patients (H1: critical, severe, and moderate disease), ambulatory patients in convalescence (O1) exhibited elevated serum IL-24 levels (*p* < 0.05). Cytokine levels exhibit pronounced temporal dynamics during infections, which are highly relevant for both diagnostic interpretation and understanding of pathophysiological processes [24]. Consequently, it is important to note that in this comparison, blood sampling in convalescent patients after a mild disease course (O1: 38 (36–46) days after symptom onset) was performed considerably later than in hospitalized patients with critical, severe, or moderate disease (H1: 11 (5–15) days after symptom onset), with the difference being statistically significant (*p* < 0.001). This is consistent with previous data describing IL-24 as one of the least abundantly expressed proteins in hospitalized COVID-19 patients. However, in those studies, neither disease severity nor the timing of blood sampling relative to symptom onset was differentiated or taken into account [25].

The role of IL-24 in host defense against viral infections remains incompletely understood. Limited data, primarily from in vitro experiments and murine models, are available in the context of influenza infection. In vitro, infection of human lung adenocarcinoma cells (A549) with Influenza A virus induced a time-dependent upregulation of IL-24 mRNA and protein expression. Pretreatment with recombinant IL-24 effectively suppressed viral plaque formation, reduced synthesis of the nonstructural protein 1 (NS1), and enhanced interferon (IFN)-induced antiviral signaling, whereas IL-24 knockdown increased viral replication [18]. In vivo, murine models of H5N8 and H5N1 influenza infection demonstrated marked upregulation of IL-24 during the course of infection. This time-dependent expression correlated with both antiviral immune responses and viral pathogenicity across influenza subtypes [16]. The underlying pathophysiological mechanisms remain largely unclear; however, in vitro studies suggest that IL-24 exerts antiviral effects partly via induction of apoptosis (Caspase-3 activation) and modulation of TLR3-mediated signaling, independent of type I interferons. Downregulation of the antiapoptotic proteins Bcl-2, Mcl-1, Bax, and Bcl-xL further augments this antiviral activity [17,18,20].

SARS-CoV-2 can induce apoptosis in immune and organ cells through various mechanisms. Activation of Caspase-3 and Caspase-8, as well as overexpression of Bax and Bak (proapoptotic proteins of the Bcl-2 family), have been demonstrated in T cells from COVID-19 patients. This leads to enhanced T-cell apoptosis and is associated with lymphopenia and severe disease progression [26,27]. The increase in T-cell apoptosis primarily affects CD4+ T cells, whose loss also impairs CD8+ T-cell function (termed “CD8 helplessness”) [26]. Structural and accessory SARS-CoV-2 proteins (e.g., M protein and ORF3a) can directly trigger apoptosis through intrinsic and extrinsic signaling pathways. The M protein stabilizes proapoptotic factors (e.g., BOK), promotes their mitochondrial translocation, and inhibits antiapoptotic signaling [28,29]. ORF3a activates Caspase-8 and facilitates mitochondrial release of cytochrome c, resulting in cellular apoptosis [29]. Furthermore, the rate of programmed CD4+ T-cell apoptosis during the acute disease phase has been linked to the development of long COVID, with patients exhibiting high proportions of apoptotic CD4+ T cells showing increased risk for persistent sequelae [27]. Beyond immune cells, lung epithelial and endothelial cells are also affected, contributing significantly to organ pathology and severe inflammatory responses [29,30]. Currently, systematic investigations of IL-24′s role in SARS-CoV-2 infection remain lacking.

Moreover, we observed that decreased IL-24 levels correlated with increased liver parameters, lymphocyte counts. This correlation suggests that in a hyperinflammatory state—characterized by elevated cytokines such as IL-6, IL-1ß, and TNF-α—IL-24 production may be disrupted or insufficient to counterbalance the inflammatory cascade [31,32]. Additionally, reduced liver function could impact IL-24 synthesis or regulation, compounding its deficiency in severe COVID-19 cases. Given IL-24′s potential modulatory effect on IFN-γ expression, this deficiency might impair antiviral cytokine responses, reducing the immune system’s ability to clear the virus effectively and increasing the likelihood of tissue damage [18,33].

Another noteworthy aspect of IL-24′s function is its involvement in allergic and autoimmune responses, as elevated anti-IL24-IgE/IgG antibodies have been linked to allergic reactions following SARS-CoV-2 vaccination [34]. This suggests that IL-24 plays a broader role in immune regulation, affecting both viral defense and inflammatory or allergic responses. Furthermore it is hypothesized that IL-24 exerts a modulatory effect on IFN-γ expression both in vitro and in vivo and thereby influences the antiviral activity of IFN-γ [35]. Additionally, future studies should investigate whether IL-24 also influences TLR7 pathways, as activation of TLR3 and TLR7 has been observed in SARS-CoV-2 infection [36]. These pathways may drive the production of additional cytokines (e.g., IL-1α, IL-1ß, IL-4, IL-6, IFN-γ, and IFN-β), which could have downstream effects on disease severity and the risk of developing conditions like ARDS [35].

While our study provides valuable insights into the relationship between IL-24 serum levels and COVID-19 disease severity, several limitations must be acknowledged. First, the observational design of the study prevents establishing a causal relationship between IL-24 levels and disease progression. Although we found correlations between IL-24 levels and disease severity, several limitations must be acknowledged. A further constraint is the unequal time interval between symptom onset and blood sampling in ambulatory convalescent patients versus hospitalized patients with critical, severe, or moderate disease, which may confound the interpretation of IL-24 dynamics across disease severity groups. Additionally, further experimental research, including clinical trials, would be needed to confirm whether modulating IL-24 levels could influence disease outcomes. Second, the sample size, particularly in the severe and critical COVID-19 groups, may limit the generalizability of our findings. A larger cohort would provide more robust data and help verify whether the observed trends hold true across diverse populations. Additionally, since our study primarily focuses on IL-24 serum levels, we were unable to assess the local expression of IL-24 in specific tissues, such as the lungs or infected cells, where its effects on apoptosis and viral replication may be more pronounced. Third, particularly among hospitalized patients with critical or severe disease, a substantial proportion of IL-24 measurements were below the lower limit of quantification (LLOQ, 62.5 pg/mL) and were set to either 0 or LLOQ/2 (31.25 pg/mL) for statistical analysis, which may compromise the statistical validity. This may indicate potentially insufficient assay sensitivity and must be considered when interpreting the results. Fourth, while we correlated IL-24 levels with markers of kidney/liver function, we did not examine the exact mechanisms underlying the modulation of IL-24 during infection. One additional limitation is the absence of data on key inflammatory cytokines (IFN-γ, IL-4, IL-2, IL-5) and the lack of analysis of their interplay with IL-24. Future studies should explore how cytokine storms, immune dysregulation, and organ-specific factors like liver dysfunction may directly influence IL-24 synthesis and its role in antiviral responses. Another limitation is the lack of detailed longitudinal data on the progression of IL-24 levels over time in individual patients. While we observed an increase in IL-24 levels over time from symptom onset in all patient groups, a more in-depth longitudinal analysis would help clarify the temporal relationship between IL-24 fluctuations and disease progression.

## 4. Materials and Methods

### 4.1. Study Subjects and Samples

In this prospective cross-sectional study, 41 COVID-19 patients (88 blood samples) were enrolled, with SARS-CoV-2 infection confirmed via nasopharyngeal swab polymerase chain reaction (PCR) testing. The study was conducted at Münster University Hospital and Marien-Hospital Steinfurt in Germany from February 2020 to February 2021. Laboratory data, along with medical histories, were extracted from patient records. COVID-19 severity was categorized based on World Health Organization (WHO) criteria: critical (requiring life-sustaining treatment, evidence of acute respiratory distress syndrome (ARDS), sepsis, septic shock; n = 11), severe (evidence of oxygenation failure, pneumonia signs, respiratory distress; n = 9), or moderate (absence of severe/critical disease progression signs; n = 8). ARDS diagnosis followed the Berlin definition [2]. Additionally, 13 individuals with mild SARS-CoV-2 infection, not requiring hospitalization, were included. Blood sampling occurred during the acute and convalescent phase. The first blood draw in hospitalized patients was performed on the day of hospital admission or inpatient transfer from other hospitals (H1). Subsequent blood draws were obtained on days 3 (H2), 8 (H3), 14 (H4), and 21 (H5) of hospitalization. Four to five months after discharge, a subset of hospitalized patients was seen in our outpatient clinic for an additional follow-up blood draw (O2). Patients with a mild disease course (no need for hospitalization) were seen once during convalescence in our outpatient clinic for blood sampling (O1, see Figure 4). The timing of each blood draw is presented relative to the onset of symptoms of SARS-CoV-2 infection. Due to differences in the length of hospital stay, some patients had single blood draws, whereas others had serial sampling during hospitalization.

The study adhered to the principles of the Declaration of Helsinki (1975, amended in 1983) and received approval from the Ethics Committee of the University of Münster (AZ 2020-220-f-S) for both study centers: Münster University Hospital and Marien-Hospital Steinfurt.

### 4.2. Blood Collection and Sample Processing

Blood samples were collected via venipuncture using Safety-Multifly needles (Sarstedt, Nümbrecht, Germany) directly into appropriate collection tubes. Sample transport to the laboratory occurred within 30 min at room temperature, followed by immediate sample processing. Hematological parameters: Complete blood count and differential blood count (leukocyte count, lymphocyte count, NLR) were determined from EDTA K3E/2.7 mL tubes (Sarstedt) and analyzed by flow cytometry immediately after collection using a Sysmex XN-9100 hematology analyzer (Sysmex Corporation, Kobe, Japan) [37]. Clinical chemistry parameters: Determination of bilirubin (colorimetric assay), AST, ALT, gamma-GT (UV/VIS photometry), CRP (turbidimetric immunoassay, TIA), albumin (UV/VIS photometry), and ferritin (electrochemiluminescence immunoassay, ECLIA) was performed from Li-heparin gel tubes LH/7.5 mL (Sarstedt) using a cobas c 702/e 801 cobas pro analyzer (Roche Diagnostics, Mannheim, Germany) according to standard protocols within 2–4 h after collection [38]. IL-24 determination: For IL-24 measurement, EDTA plasma was collected in EDTA K3E/7.5 mL tubes (Sarstedt). Tubes were centrifuged within 30 min after collection at 3000× *g* for 10 min at room temperature. The obtained plasma was cryopreserved at −80 °C until batch analysis. IL-24 concentrations were determined using the IL24 Human DuoSet ELISA Kit (R&D Systems, Minneapolis, MN, USA) according to the manufacturer’s protocol [39]. The assay has a range of 62.5–4000 pg/mL, and the sensitivity for IL24 was not specified. Samples exceeding the upper limit of the test were subjected to a re-measurement following a 10-fold dilution in Calibrator Diluent RD6-10 reagent, as per the manufacturer’s specifications. For values below the lower limit of quantification (LLOQ, 62.5 pg/mL), two approaches were used: in one analysis, these values were set to zero; in a comparative analysis, they were assigned a value of 31.25 pg/mL. All samples were processed according to standardized Standard Operating Procedures (SOPs) to ensure sample integrity and minimize preanalytical variability.

### 4.3. Statistical Analysis

Descriptive data are presented as median with interquartile range (IQR, 25th–75th percentile) for continuous variables due to non-normal distribution, and as absolute frequencies with percentages for categorical variables. For categorical variables, either Fisher’s exact test or chi-square test was employed, and the results are presented as absolute numbers and percentages. Continuous variables were compared using the Mann–Whitney U test (Wilcoxon test), indicating the median and interquartile range. A Kruskal–Wallis test was utilized for comparisons involving more than two groups. Subgroup analysis involved a Bonferroni correction post hoc test (Levene test for equal variance) or Games–Howell test (unequal variance). To investigate the association between IL-24 and other laboratory parameters, a linear mixed-effects model (LMM) was used, including a random intercept for patient ID to account for within-subject correlation due to repeated measurements. All tests were two-tailed, and statistical significance was interpreted as *p* < 0.05. Statistical analyses were performed using SPSS 26 (IBM, Armonk, NY, USA), Graphpad Prism software (Version 9, Graphpad Software, La Jolla, CA, USA), R package and Rstudio Version 4.5.1 (RStudio Team (2015). RStudio: Integrated Development for R. RStudio, Inc., Boston, MA, USA, http://www.rstudio.com/ (accessed on 13 August 2025).

## 5. Conclusions

In summary, our findings propose that IL-24 could serve as a prognostic marker, owing to its elevation in milder disease courses and correlation with better outcomes. Targeting IL-24 pathways therapeutically could help enhance viral clearance and mitigate severe COVID-19 outcomes, particularly among patients with low IL-24 levels or those at elevated risk for ARDS. Further research is essential to elucidate IL-24′s specific antiviral effects against SARS-CoV-2 and its interactions with key immune pathways in COVID-19 pathophysiology.

## Figures and Tables

**Figure 1 ijms-26-08403-f001:**
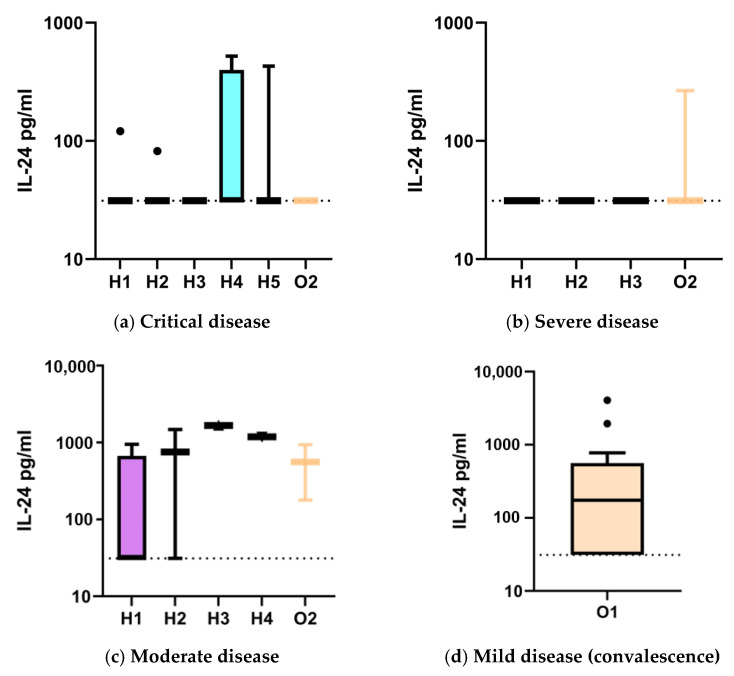
IL-24 levels (pg/mL) in hospitalized COVID-19 patients according to disease course—critical (**a**), severe (**b**), and moderate (**c**)—with blood samples obtained at hospital admission or inpatient transfer (H1), on day 3 (H2), day 8 (H3), day 14 (H4), and day 21 (H5). Following hospital discharge, an additional blood sample was obtained in some patients 4 to 5 months later (O2). IL-24 levels in COVID-19 patients with a mild disease course, sampled during convalescence (O1), are shown in panel (**d**). For IL-24 values below the lower limit of quantification (LLOQ, 62.5 pg/mL), these values were set to 31.25 pg/mL, shown as a dashed line. Values are presented as Tukey boxplots, with the box representing the interquartile range (IQR), the horizontal line indicating the median, and whiskers extending to 1.5 × IQR; outliers are shown as individual points. The y-axis is scaled logarithmically (log10).

**Figure 2 ijms-26-08403-f002:**
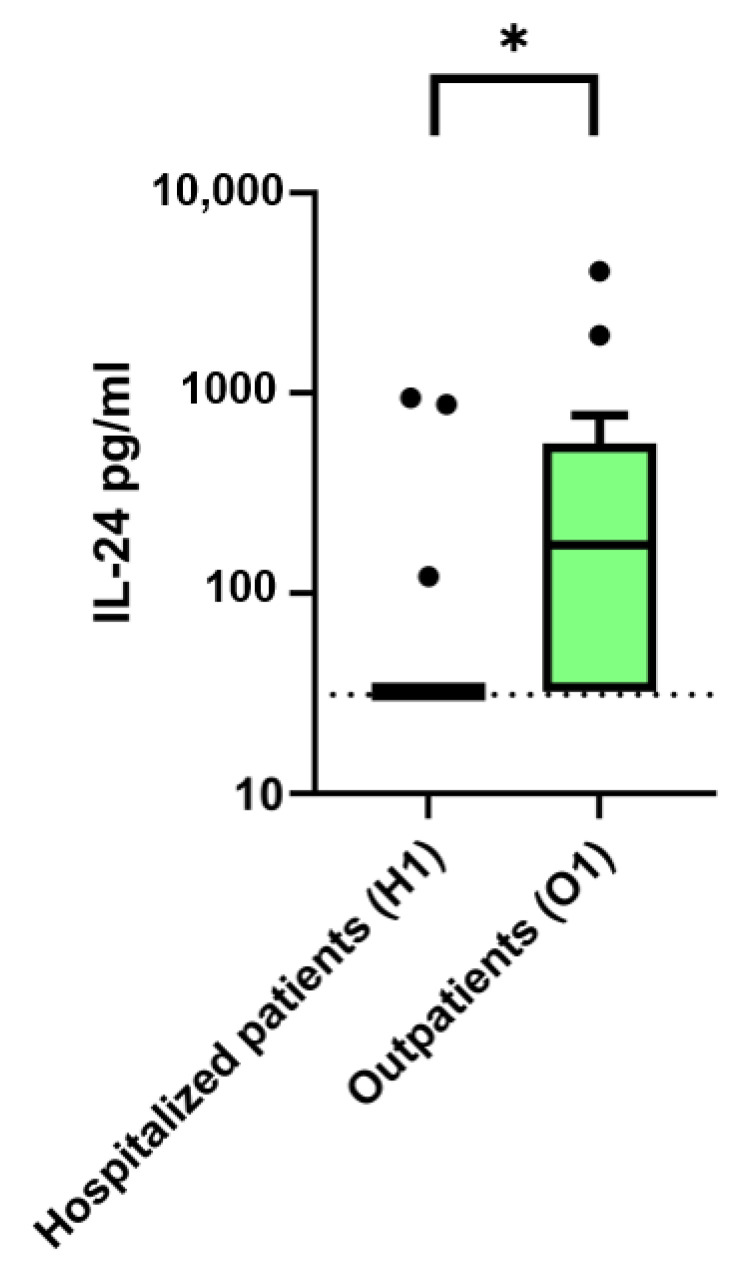
IL-24 serum levels (pg/mL) in COVID-19 patients according to disease severity. Hospitalized patients with critical, severe, and moderate disease courses at the time of hospital admission or inpatient transfer (H1) are compared to outpatients after mild disease course during convalescence (O1). For IL-24 values below the lower limit of quantification (LLOQ, 62.5 pg/mL), values were set to half the LLOQ (31.25 pg/mL), shown as a dashed line. Values are presented as Tukey boxplots, where the box represents the interquartile range (IQR), the horizontal line indicates the median, whiskers extend to 1.5 × IQR, and outliers are shown as individual points. The *y*-axis is displayed on a logarithmic scale (log10). IL-24, Interleukin 24. Continuous variables were compared using the Mann–Whitney U test (Wilcoxon test), * *p* < 0.05.

**Figure 3 ijms-26-08403-f003:**
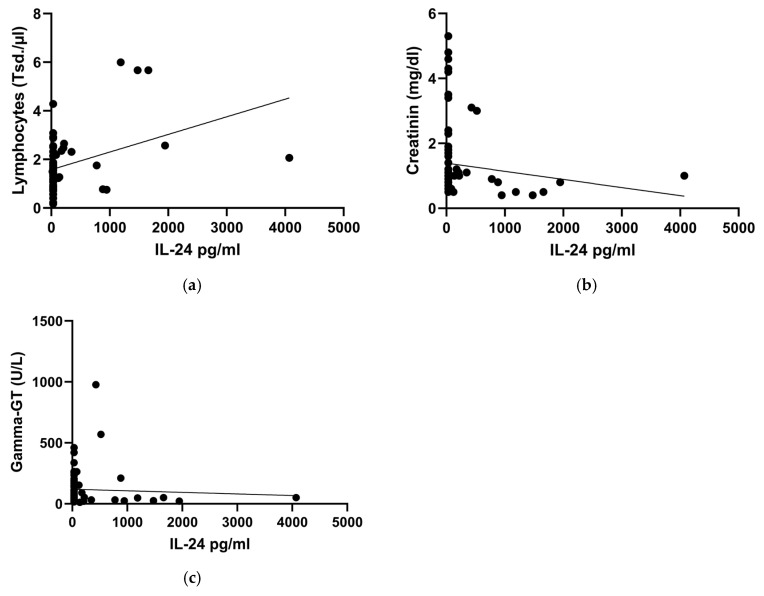
Correlations between Interleukin-24 (IL-24) serum levels and lymphocyte counts (**a**), creatinine (**b**) and gamma-GT (**c**). Regression analysis plot (linear mixed-effects model): absolute lymphocyte count *p* = 0.026, creatinine *p* = 0.043, gamma-GT *p* = 0.002. These correlations include IL-24 values from hospitalized COVID-19 patients regardless of disease severity (critical, severe, moderate) and outpatients after convalescence, with the respective laboratory parameters. For IL-24 values below the lower limit of quantification (LLOQ, 62.5 pg/mL), values were set to half the LLOQ (31.25 pg/mL).

**Figure 4 ijms-26-08403-f004:**
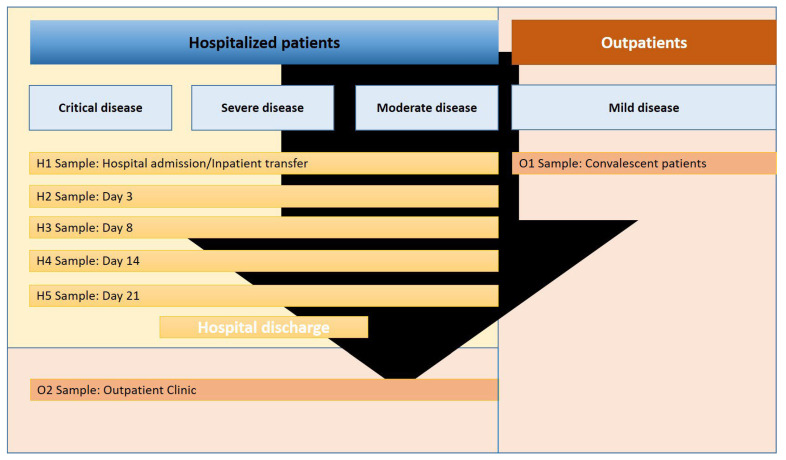
Study flowchart depicting blood sampling timepoints in COVID-19 patients. Hospitalized COVID-19 patients with critical, severe, or moderate disease courses underwent blood sampling at hospital admission or inpatient transfer (H1, first blood draw). Subsequent samples were collected on days 3 (H2), 8 (H3), 14 (H4), and 21 (H5). Some patients were followed up with an additional blood draw in the outpatient clinic approximately 4–5 months after symptom onset (O2). Additionally, COVID-19 patients with a mild disease course who did not require hospitalization were enrolled and sampled once during convalescence (O1).

**Table 1 ijms-26-08403-t001:** Cohort Characteristics: Differences of continuous variables were assessed using the Kruskal–Wallis test, categorical variables were compared using the Fisher-Freeman-Halton exact test. IQR, interquartile range; BMI, Body Mass Index.

Patients n = 41 (Samples n = 88)		Hospitalized			Outpatients	*p*-Value
		Critical Disease	Severe Disease	Moderate Disease	Mild Disease	
		Oxygen therapy		No oxygen therapy		
**Patient** **characteristics**	Patients, No.	11	9	8	13	
	Samples, No.	37	24	14	13	
	Sex, male, No. (%)	10 (91)	8 (89)	6 (75)	13 (100)	0.19
	Age, median (min–max)	59 (41–72)	53 (44–76)	50 (30–66)	47 (31–70)	0.314
	BMI, median (IQR)	26 (23–29)	24 (23–26)	25 (22–27)	29 (25–31)	0.057
	Time symptom onset to sample (1. Sample)	12 (11–17)	11 (3–17)	8 (2–8)	38 (36–46)	<0.01
**Pre-existing conditions**	Diabetes (%)	1 (9)	0 (0)	1 (13)	0 (0)	
	Chronic inflammatory bowel disease (%)	1 (9)	1 (11)	0 (0)	0 (0)	
	Coronary heart disease (%)	2 (18)	1 (11)	1 (13)	0 (0)	
	Respiratory disease (%)	1 (9)	0 (0)	1 (13)	0 (0)	
	Neoplasm (%)	1 (9)	0 (0)	0 (0)	0 (0)	
	Arterial hypertension (%)	0 (0)	0 (0)	0 (0)	0 (0)	

**Table 2 ijms-26-08403-t002:** Labor values of COVID-19 patients based on the course of the disease. Differences of continuous variables were assessed using the Kruskal–Wallis test. IQR, interquartile range; ALT, alanine aminotransferase; AST, aspartate transaminase; CRP, C-reactive protein, NLR, Neutrophil-Lymphocyte Ratio.

Patients n = 41 (1. Sample)	Critical Disease	Severe Disease	Moderate Disease	Mild Disease	*p*-Value
**Leukocytes** (10^9^/L), median (IQR)	8.7 (5.3–12.7)	5.4 (3.4–7.6)	4.1 (3.5–5.2)	5.6 (5.1–7.4)	0.014
**Lymphocytes** (rel., %), median (IQR)	11.3 (7.2–14.5)	21.6 (18.5–31.5)	25.8 (19.3–32.4)	32.5 (28.1–38)	0.001
**Lymphocytes** (abs. 10^9^/L), median (IQR)	1.5 (0.7–1.7)	1.2 (0.9–1.6)	0.96 (0.76–1.3)	2.3 (1.4–2.5)	0.013
**NLR**	6.7 (5.3–11.4)	2.8 (1.5–2.9)	2.2 (1.8–3.8)	1.7 (1.4–2.2)	<0.001
**Bilirubin** (mg/dL), median (IQR)	0.5 (0.3–1.3)	0.5 (0.5–0.6)	0.5 (0.3–0.8)	0.4 (0.4–0.5)	0.471
**AST** (U/L), median (IQR)	81 (59–134)	41 (32–62)	29 (25–45)	32 (25–42)	0.001
**ALT** (U/L), median (IQR)	66 (27–114)	34 (29–60)	22 (18–47)	29 (26–50)	0.204
**Gamma-GT** (U/L), median (IQR)	97 (55–146)	54 (30–60)	41 (33–124)	32 (21–52)	0.014
**CRP** (mg/dL), median (IQR)	14 (6–30)	5.2 (2.4–9.4)	2 (0.8–3.3)	0.5 (0.5–0.5)	<0.001
**Albumin** (g/dL), median (IQR)	2 (1–2.6)	3.5 (3.1–4.3)	4.0 (3.7–4.2)	4.6 (4.5–4.8)	<0.001
**Ferritin** (µg/L), median (IQR)	999 (646–1937)	779 (362–938)	292 (191–557)	248 (96–310)	<0.001

**Table 3 ijms-26-08403-t003:** L-24 serum levels in COVID-19 patients according to disease severity (critical, severe, moderate, mild). Blood samples were obtained at hospital admission or inpatient transfer (H1), on day 3 (H2), day 8 (H3), day 14 (H4), and day 21 (H5) of hospitalization. Additional outpatient samples after discharge (O2) and convalescent samples from patients with mild disease (O1) were also collected. For IL-24 values below the lower limit of quantification (LLOQ, 62.5 pg/mL), two approaches were used: in one analysis, these values were set to zero *; in a comparative analysis, they were assigned a value of 31.25 pg/mL ^#^. IL-24, Interleukin.

Patients n = 41 (88 Samples)	Critical Disease	Severe Disease	Moderate Disease	Mild Disease
**Hospitalized patients**				
**H1 Sample**	**11**	**9**	**8**	
**IL-24 *** (pg/mL), median (IQR)	0 (0–0)	0 (0–0)	0 (0–661)	
**IL-24 ^#^** (pg/mL), median (IQR)	31.3 (31.3–31.3)	31.3 (31.3–31.3)	31.3 (31.3–661)	
**IL-24 values above LLOQ**, n (%)	1 (9)	0 (0)	2 (25)	
**Time** symptom onset to sample (days)	12 (11–17)	11 (3–17)	8 (2–8)	
**H2 Sample**	**9**	**7**	**2**	
**IL-24 *** (pg/mL), median (IQR)	0 (0–0)	0 (0–0)	754.1 (0)	
**IL-24 ^#^** (pg/mL), median (IQR)	31.3 (31.3–31.3)	31.3 (31.3–31.3)	754.1 (31.3)	
**IL-24 values above LLOQ**, n (%)	1 (11)	0 (0)	1 (50)	
**Time** symptom onset to sample (days)	17 (14–21)	14 (4–22)	4 (3)	
**H3 Sample**	**7**	**5**	**1**	
**IL-24 *** (pg/mL), median (IQR)	0 (0–0)	0 (0–0)	1660	
**IL-24 ^#^** (pg/mL), median (IQR)	31.3 (31.3–31.3)	31.3 (31.3–31.3)	1660	
**IL-24 values above LLOQ**, n (%)	0 (0)	0 (0)	1 (100)	
**Time** symptom onset to sample (days)	22 (17–26)	13 (11–26)	8	
**H4 Sample**	**4**		**1**	
**IL-24 *** (pg/mL), median (IQR)	0 (0–398.6)		1188	
**IL-24 ^#^** (pg/mL), median (IQR)	31.3 (31.3–398.6))		1188	
**IL-24 values above LLOQ**, n (%)	1 (25)		1 (100)	
**Time** symptom onset to sample (days)	31 (27–37)		10	
**H5 Sample**	**3**			
**IL-24 *** (pg/mL), median (IQR)	0 (0–0)			
**IL-24 ^#^** (pg/mL), median (IQR)	31.3 (31.3–31.3)			
**IL-24 values above LLOQ**, n (%)	1 (33)			
**Time** symptom onset to sample (days)	46 (27)			
**Outpatients**				
**O1 Sample**				**13**
**IL-24 *** (pg/mL), median (IQR)				174 (0–561)
**IL-24 ^#^** (pg/mL), median (IQR)				174 (31.3–561)
**IL-24 values above LLOQ**, n (%)				8 (62)
**Time** symptom onset to sample (days)				38 (36–46)
**O2 Sample**	**3**	**3**	**2**	
**IL-24 * (pg/mL), median (IQR)**	0 (0–0)	0 (0)	558 (178)	
**IL-24 ^#^ (pg/mL), median (IQR)**	31.3 (31.3–31.)	31.3 (31.3)	558 (178)	
**IL-24 values above LLOQ**, n (%)	0 (0)	1	2	
**Time** symptom onset to sample (days)	140 (126)	126 (122)	137 (129)	

## Data Availability

The data are not publicly available due to privacy/ethical restrictions.

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
