# Peer review of "IL-24 in COVID-19 Patients: Correlations with Disease Progression"

_ijms, 2025, doi:10.3390/ijms26178403_

Round 1
Reviewer 1 Report
Comments and Suggestions for Authors
Vollenberg and co-workers aimed to explore IL-24 levels in COVID-19 disease, in several groups of patients, classified according to disease severity. They measured IL-24 levels in 41 patient, and there were 88 IL-24 measures (which correspond to one ELISA plate, including samples and standard curve). As presented in the manuscript, the study design is unclear and confusing. There is a large difference in time sampling beetween the groups. While in hospitalized patients first sample is taken during acute phase of illness, in mild disease sampling was performed after more than 30 days, when disease was almost resolved. It is obvious that there are several samples per same participant, but sampling time and scheme is unclear. In general, study design is weak, and conclusions driven must be taken with high reserve. The authors state that they correlated IL-24 levels with inflammatory markers, but no such analysis was shown in this work. The methodology is incomplete and scarce, ethical approval was obtained only from one hospital, not the other one participating. Statistical analysis is also questionable, as in some groups measured IL-24 levels were 0, which means actually below LoQ or immeasurable. Also, some preexisting conditions were listed, despite they were present in isolated cases or even not present at all, and some statistic was performed, even though it is not relevant at all.
The more detailed comments are given in the manuscript itself. Please find attached file.

Author Response
Thank you for the thorough engagement with the manuscript. We have addressed and revised all the corresponding points.
- To clearly present the study design to the reader, we have expanded the laboratory sample analysis to include a breakdown of additional blood sampling time points and have comprehensively supplemented this in the Methods and Results sections including figures/tables. The different sampling periods for hospitalized COVID-19 patients are now clearly visible. The limitation of varying intervals between blood sampling and symptom onset depending on disease severity (particularly in ambulatory patients after mild disease course) has been more clearly elaborated and discussed.
- Due to the study design and particularly the small sample size, the results must be interpreted with caution. We have adjusted the discussion accordingly.
- We have revised and adapted the section on correlations between IL-24 and other measured values.
- We have revised the Methods section and expanded it to include a detailed description of the methods used.
- Additionally, the Methods, Results, and Discussion sections have been expanded to provide a detailed presentation of values below the LLOQ (62.5 pg/ml). These values were set to zero and to LLOQ/2 (31.25 pg/ml), and all subsequent analyses were performed using LLOQ/2. We have also thoroughly addressed the potential limitations associated with this approach. Table 3 now clearly shows how many patients in each group had values below the LLOQ.
- We have revised the section on comorbidities accordingly.

Reviewer 2 Report
Comments and Suggestions for Authors
The paper shows valuable data on IL-24 levels in COVID-19 patients with various disease severity and with time progression. The prospective study unveiled IL-24 correlations with severity and clinical blood markers. However, in the text authors abuse the construction of conclusions without relying on the data obtained in the work, especially in Discussion section. Additionally statistical analysis and data processing contain serious issues that should be addressed, after which the correlations that were found by authors should be reevaluated.
Below are the specific issues, questions and comments.
1. Abstract contain text "Nachhalten der Biopsien, bei 17 Nachweiseiner Helicobacter-Pylori-Infektion entsprechender Helicobacter-Eradi-18 kationstherapie." that is probably added by mistake.
2. Statement "suggesting a protective role in modulating immune responses and promoting antiviral apoptosis" in Abstract is not supported by results in paper or literature review, therefore could not be added.
3. Statement "immunocompromised patients often suffer from insufficient immunity even after vaccination" is generous and must be specified. Respective references may reinforce point.
4. Question to statement "Strikingly, it could be shown that IL-24 is the most upregulated differentially expressed genes in the lungs of H5 avian influenza virus-infected mice". Mice and human cytokine responses are not equivalent, they are different. Implicitly authors declare that result obtained for animals could be transferred for humans. Could authors provide the supporting evidence that mice and human pathology in terms of IL-24 responses is resembled for influenza infection?
5. In line "IFN-ß" it seems that Greek beta symbol is replaced with German Eszett symbol.
6. The time of blood collection for ambulant patients significantly differs from one for hospitalized. Authors should take into account that cytokine response is not long-lasting and adjust interpretation of results accordingly.
7. Table 1 contain typo in header: "Milde" -> "Mild".
8. Data in Figure 1 is presented in poor way. Obvious outliers should be removed or put in separate panel/view so that major number of data points will be visible for the reader. Currently it is hard to make sure that results described in text are confirmed by data. Moreover authors do not provide raw data, what makes results less trustworthy. Panels (a) and (b) must be changed: add subpanels to show major set of points or use log-scale for y-axis. Panels (e) and (f) have bad choice of limits for y-axis, it must be set to lower values so that data points will be visible. Additionally labels for these panels are located at previous page separately: this must be fixed so that labels and panels will be at the same page.
9. The ELISA kit IL24 DuoSet from R&D Systems allows to measure concentration of IL-24 in range 62.5-4000 pg/ml. Taking into account that limit of quantification is 62.5 pg/ml (lower limit of range) it is alarming that authors report values of concentrations below 62.5 pg/ml. It is impossible to reliably measure lower concentrations. All the data below 62.5 pg/ml should not be considered as 0 pg/ml in correlation and statistical analysis.
10. Figure 1 c, d, e, f and Figure 2 a, b contain trend lines that were calculated data obtained below limit of quantification of ELISA kit. This is wrong and analysis should take into account uncertainty for points that lay below limit of quantification.
11. Following statements in Discussion section have no supporting evidence from author's data or literature, it should be removed.
"Such a mechanism may also apply to COVID-19, where IL-24 could support the elimination of SARS-CoV-2-infected cells by enhancing apoptotic pathways and limiting viral replication."
"This deficiency could indicate an impaired apoptotic response, which may allow continued viral replication and immune dysregulation, potentially leading to cytokine storms and severe inflammation."
"...suggesting that IL-24 may play a protective role in preventing severe pulmonary complications."
"Patients with low IL-24 levels may thus be at greater risk of developing severe respiratory symptoms, as the lack of IL-24-mediated apoptosis might hinder the clearance of infected cells, allowing viral persistence and exaggerated immune responses that damage lung tissue." - no risk probabilities calculation provided; no viral load data provided (in principle, this could have been foreseen in the study design at the beginning).
Paragraph in lines 167-179 in general not supported by data of this paper. I suggest author to not speculate on implications without solid evidence provided.
On the other hand appropriate and sound discussion of results is given in paragraph in lines 180-188.
12. In line 182 again Greek beta is replaced with Eszett.
13. In line 187 it seems "contain" should be replaced with "clear".
14. Authors provide versatile discussion of study limitations that shows experience and erudition in the field. I suggest to add one more limitation: lack of mentioning the data for key inflammatory cytokines (IFNg, IL-4, IL-2, IL-5) and analysis of their interplay with IL-24. Discussion of more general view on cytokine response during and post COVID-19 could increase the value of paper.
15. No reference is given to manufacturer of PCR kit.
16. No reference is given to manufacturer of kits for laboratory analyses of blood.
17. No measurement instrument information is given for ELISA, PCR and other markers.
18. In Conclusion session following statement is not supported by evidence and should be removed: "suggests it may assist in viral elimination through apoptosis, helping contain the infection and prevent progression to severe disease"
19. Unavailable data and poor presentation on figures makes it hard to trust conclusions.
Author Response
We thank the reviewer for his good advice to improve the publication. We have addressed all the points raised below.

Reviewer 3 Report
Comments and Suggestions for Authors
The research objective of this manuscript is to investigate the association between serum IL-24 levels and the severity of disease in COVID-19 patients. However, the abstract section (line 17) includes irrelevant content related to Helicobacter pylori.
Overall, the manuscript demonstrates a certain degree of scientific foundation and innovative potential. However, it exhibits numerous deficiencies in academic rigor and experimental methodology. The literature review would benefit from incorporating additional content on the role of IL-24 in respiratory virus infections and the current research progress regarding IL-24 in COVID-19, which would enhance the study’s significance. The discussion section lacks an in-depth analysis of the results, such as exploring the potential mechanisms underlying the reduction in IL-24 levels and whether these are related to immune suppression or cytokine storms induced by SARS-CoV-2 infection. Additionally, the discussion of methodological limitations is insufficient, particularly regarding the small sample size, heterogeneity in the patient cohort, and the rationale for the selection of sample collection time points. Furthermore, the manuscript primarily relies on the IL-24 ELISA assay for detection, but the measurement of zero in multiple samples raises concerns about the sensitivity and accuracy of this method.
Author Response
Thank you for this important feedback on our manuscript. We have removed the incorrect reference to Helicobacter pylori. To improve understanding of the results, we have further broken down the serial blood samplings by collection time point and described the study design in more detail in the Methods section, including the addition of a study flow chart. Additionally, the Methods, Results, and Discussion sections have been expanded to provide a detailed presentation of values below the LLOQ (62.5 pg/ml). These values were set to zero and to LLOQ/2 (31.25 pg/ml), and all subsequent analyses were performed using LLOQ/2. We have also thoroughly addressed the potential limitations associated with this approach. Table 3 now clearly shows how many patients in each group had values below the LLOQ. Regarding IL-24 and respiratory infections, there is still insufficient evidence, and the data—particularly under in vivo conditions—remain the subject of ongoing research. We have expanded on these points in the Discussion.

Round 2
Reviewer 2 Report
Comments and Suggestions for Authors
The paper shows valuable data on IL-24 levels in COVID-19 patients with various disease severity and with time progression. The prospective study unveiled IL-24 correlations with severity and clinical blood markers.